# Sequential Harmful Shift Detection Without Labels

**Salim I. Amoukou** * **Tom Bewley** **Saumitra Mishra**
**Freddy Lecue** **Daniele Magazzeni** **Manuela Veloso**
J.P. Morgan AI Research

## Abstract

We introduce a novel approach for detecting distribution shifts that negatively impact the performance of machine learning models in continuous production environments, which requires no access to ground truth data labels. It builds upon the work of Podkopaev and Ramdas [2022], who address scenarios where labels are available for tracking model errors over time. Our solution extends this framework to work in the absence of labels, by employing a proxy for the true error. This proxy is derived using the predictions of a trained error estimator. Experiments show that our method has high power and false alarm control under various distribution shifts, including covariate and label shifts and natural shifts over geography and time.

## 1 Introduction

When deploying a machine learning model in production, it is common to encounter changes in the data distribution, such as shifts in covariates [Shimodaira, 2000], labels [Saerens et al., 2002, Lipton et al., 2018] or concepts [Gonçalves Jr et al., 2014]. Many methods exist for detecting such distribution shifts. However, a distinct but equally important challenge is assessing whether a shift has a harmful impact on the prediction error of a given model, which may necessitate interventions such as ceasing production or retraining the model. Not all distribution shifts are harmful, but traditional methods for shift detection are unable to distinguish harmful and benign shifts.

While some approaches address the specific issue of performance shift, most require access to ground truth data labels in the production environment [Gama et al., 2013, 2014, Bayram et al., 2022]. In scenarios where predictions concern future outcomes, such as medical diagnosis or credit scoring, immediate access to labels in production is not feasible. This work focuses on the challenge of detecting harmful distribution shifts — those that increase model error in production — without requiring access to labels. As Trivedi et al. [2023] note, current methods for harmful shift detection without labels rely on disparate heuristics, often lacking a solid theoretical foundation. Such methods include proxies based on aggregate dataset-level statistics [Deng and Zheng, 2021], optimal transport mappings between training and production distributions [Koebler et al., 2023], and model-specific metrics such as input margins [Mouton et al., 2023], perturbation-sensitivity [Ng et al., 2023], disagreement-metrics [Chen et al., 2023, Ginsberg et al., 2022], and prediction confidence [Guillory et al., 2021, Garg et al., 2022]. While such methods may be practically effective in certain contexts, they rely on assumptions and correlations that do not hold universally, so can provide no guarantees.

Furthermore, conventional methods rely on two-sample or batch testing, which involves comparing the statistical properties of a production dataset with those of a control sample. These methods have inherent limitations, as the sample size is prespecified. This is a problem because the necessary amount of data to detect any given shift is unknown beforehand. Furthermore, in real-world scenarios, data typically arrive sequentially over time and shifts may occur either suddenly or gradually. In such scenarios, it may be desirable to detect harmful shifts as early as possible. Batch testing is ill-suited to the sequential context [Maharaj et al., 2023], as it does not accommodate the collection of additional data for retesting without adjusting for multiple testing, leading to diminished power.

---

*Correspondence to: Salim I. Amoukou <salim.ibrahimamoukou@jpmorgan.com>

The most principled and relevant work to our problem is that of Podkopaev and Ramdas [2022], which tackles the problem of sequential harmful shift detection with false alarm control but assumes the availability of ground truth labels in production. Our work builds on the foundation established by Podkopaev and Ramdas [2022], extending the methodology to detect harmful shifts in unlabeled production data while effectively managing false alarms.

Our approach leverages a secondary model to estimate the errors of the primary model. While learning such a model might seem challenging at first, consider a situation where the primary model performs well overall but struggles with specific data subgroups. Sagawa et al. [2019] demonstrate that this phenomenon can occur in natural distributions. In such cases, learning to predict "error given X" might be easier than the primary task of predicting "Y given X", because the error estimator only needs to identify those subgroups where the primary model struggles. This approach has shown promise in recent studies [Zrnic and Candès, 2024, Amoukou and Brunel, 2023]. More generally, Zrnic and Candès [2024] note that predicting the magnitude of the error, rather than its direction, is often easier. Furthermore, our approach is based on estimating the proportion of high-error observations over time. For this task, the error estimator does not need to be very accurate; it only needs to assign higher values to observations with higher errors. That is, the estimator only needs to correctly order most observations from low to high error, which is easier than precisely predicting the error itself. We demonstrate in Section 4.1 that even a relatively inaccurate error estimator can be effective at identifying high-error observations, and thus provides the functionality required by our framework. Although this paper uses a learned error estimator's predictions as a proxy for error, we note that any scalar function correlated with error could suffice to isolate high-error observations. For example, for a well-calibrated binary classification model, we could instead use that model's predicted probability, tracking observations with predictions near 0.5 to identify uncertain predictions.

Figure 1 gives an overview of our approach. We first fit the secondary error estimator model to predict the error of the primary model, then use labeled data to calibrate an estimated error threshold (---) that separates observations with low (●) and high (●) true error as fully as possible. We run the error estimator on all observations encountered in production and continually monitor the proportion of observations whose estimated error falls above the threshold. We raise an alarm when this exceeds the rate of high-error observations (●) in the calibration set plus a tolerance threshold $\epsilon_{\text{tol}}$ and correction terms to deal with the sequential setting and account for uncertainty in the estimates. In the example shown, this occurs at time $t = 10$.

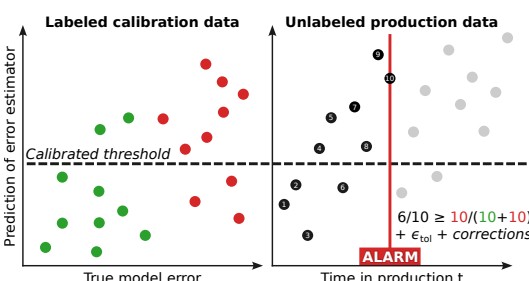

Figure 1: Overview of the proposed approach. **Left:** calibrating an estimated error threshold to separate low/high true errors. **Right:** sequentially tracking production data exceeding the threshold and raising an alarm upon a significant increase.

The rest of this paper is organized as follows. Section 2 outlines the problem definition and Section 3 provides an overview of the foundational work of Podkopaev and Ramdas [2022] as background. Section 4 is dedicated to the presentation and theoretical analysis of our sequential statistical test. Section 5 demonstrates the empirical efficacy of our method, showcasing its strong detection capabilities and controlled false alarm rates across various types of harmful shift.

## 2 Problem Definition

Let $\mathcal{X}$ and $\mathcal{Y}$ be input and label spaces, $f : \mathcal{X} \to \mathcal{Y}$ be a predictive model, and $\ell : \mathcal{Y}^2 \to \mathcal{E}$ be a measurable and bounded error function that is selected for monitoring purposes. The model's error on a specific observation $(\boldsymbol{X}, Y) \in \mathcal{X} \times \mathcal{Y}$, drawn from a joint distribution $P_{(\boldsymbol{X}, Y)} = P_{\boldsymbol{X}} P_{Y|\boldsymbol{X}}$, is represented by the random variable $E = \ell(f(\boldsymbol{X}), Y)$. The probability distribution of the error is denoted by $P_E$. As discussed above, our focus is not on detecting shifts in covariates or labels per se, but rather changes in the error distribution $P_E$. Error changes can be caused by various types of shift in the underlying joint distribution, including changes in $P_X$ while the conditional label distribution $P_{Y|\boldsymbol{X}}$ remains constant (covariate shift) or changes in $P_Y$ while $P_{\boldsymbol{X}|Y}$ remains constant (label shift).

We assume access to a dataset $\mathcal{D}_n = \{(\boldsymbol{X}_i^0, Y_i^0)\}_{i=1}^n$, sampled independently from a *source* distribution $P_{(\boldsymbol{X}, Y)}^0$. In addition, we have a sequence of data $(\boldsymbol{X}_t, Y_t)_{t \geq 1}$ drawn independently from a time-

varying distribution encountered by the model in production, $P^t_{(\boldsymbol{X},Y)}$. We model the ocurrence of a shift in production by assuming this distribution is equal to the source before some time $T \in \mathbb{N} \cup \{\infty\}$ (i.e., $P^t_{(\boldsymbol{X},Y)} = P^0_{(\boldsymbol{X},Y)}, \forall t < T$) and different thereafter (i.e., $P^t_{(\boldsymbol{X},Y)} \neq P^0_{(\boldsymbol{X},Y)}, \forall t \geq T$).

Our goal whenever there is a shift, (i.e., $T < \infty$), is to decide if this shift is harmful to the model error. To formalize this, we introduce $\theta : \mathcal{P}(\mathcal{E}) \to \mathbb{R}^+$ as a mapping from probability distributions on the error space $\mathcal{E}$ to a real-valued parameter. This mapping could, for instance, map the distribution to its mean or a certain quantile. We aim to construct a sequential test for the following pair of hypotheses:

$$H_0 : \forall t \geq 1, \ \left( \tfrac{1}{t} \sum_{k=1}^t \theta(P^k_E) \right) \leq \theta(P^0_E) + \epsilon_{\text{tol}}; \tag{1}$$

$$H_1 : \exists t \geq T : \left( \tfrac{1}{t} \sum_{k=1}^t \theta(P^k_E) \right) > \theta(P^0_E) + \epsilon_{\text{tol}}, \tag{2}$$

where $P^k_E$ denotes the error distribution at at time $k$, the *running risk* $\frac{1}{t} \sum_{k=1}^t \theta(P^k_E)$ is the average value of the error parameter up to time $t$, and $\epsilon_{\text{tol}} \geq 0$ is a tolerance level. Intuitively, $H_0$ holds if the running risk remains below that of the source distribution ($+\epsilon_{\text{tol}}$) for all time throughout production, and $H_1$ holds if this condition is violated.

**Objective.** Construct a $\alpha$-level sequential test, defined by an *alarm* function $\Phi : \cup_{k=1}^\infty \mathcal{X}^k \to \{0, 1\}$, which at time $t$ uses the first $t$ observations $\boldsymbol{X}_1, \ldots, \boldsymbol{X}_t$ to output 0 (no harmful shift so far) or 1 (harmful shift; raise an alarm) with a controlled false alarm rate and high power, i.e.,

$$\mathbb{P}_{H_0} (\exists t \geq 1 : \ \Phi(X_1, \ldots, X_t) = 1) \leq \alpha, \quad \text{and} \quad \mathbb{P}_{H_1} (\exists t \geq 1 : \ \Phi(X_1, \ldots, X_t) = 1) \approx 1. \tag{3}$$

We refer to this problem definition as *sequential harmful shift detection* (SHSD).

## 3 SHSD with Production Labels

A work closely related to ours is that of Podkopaev and Ramdas [2022], which offers a solution for scenarios where the ground truth labels of the production data are available. This method leverages confidence sequences [Darling and Robbins, 1967, Jennison and Turnbull, 1984, Johari et al., 2015, Jamieson and Jain, 2018], which are time-uniform (i.e., valid for any time) confidence intervals, allowing for the ongoing monitoring of any bounded random variable. With access to labels, it is possible to calculate the true errors on the production data over time and monitor the running risk.

Choosing the mean as the error parameter i.e. $\theta(P^k_E) = \mathbb{E}_{P^k}[E]$, Podkopaev and Ramdas [2022] use the empirical production errors $E_1 = \ell(f(\boldsymbol{X}_1), Y_1), \ldots, E_t = \ell(f(\boldsymbol{X}_t), Y_t)$, to construct a confidence sequence lower bound $\hat{L}$ for the running risk, satisfying a chosen miscoverage level $\alpha_{\text{prod}} \in (0, 1)$:

$$\mathbb{P}\Big( \forall t \geq 1, \ \big( \tfrac{1}{t} \sum_{k=1}^t \theta(P^k_E) \big) \geq \hat{L}(E_1, \ldots, E_t) \Big) \geq 1 - \alpha_{\text{prod}}. \tag{4}$$

This equation guarantees that the lower bound remains valid over time with high probability. Furthermore, given the errors on the source data $E^0_1 = \ell(f(\boldsymbol{X}^0_1), Y^0_1), \ldots, E^0_n = \ell(f(\boldsymbol{X}^0_n), Y^0_n)$, either another confidence sequence or a traditional confidence interval method [Howard et al., 2021, Waudby-Smith and Ramdas, 2020] can be used to construct a fixed-time upper confidence bound $\hat{U}$ for the mean error $\theta(P^0_E)$. For a miscoverage level $\alpha_{\text{source}} \in (0, 1)$, $\hat{U}$ satisfies the following condition:

$$\forall n \geq 1, \ \mathbb{P}\left( \theta(P^0_E) \leq \hat{U}(E^0_1, \ldots, E^0_n) \right) \geq 1 - \alpha_{\text{source}}. \tag{5}$$

An alarm is raised when the lower bound of the running risk in production exceeds the upper bound of the source error plus a tolerance $\epsilon_{\text{tol}}$. Formally, this equates to defining the function $\Phi$ as follows:

$$\Phi_m(E_1, \ldots, E_t) = \mathbb{1}\left\{ \hat{L}(E_1, \ldots, E_t) > \hat{U}(E^0_1, \ldots, E^0_n) + \epsilon_{\text{tol}} \right\}, \tag{6}$$

where the subscripted $m$ denotes that the mean is the error parameter being tracked. This methodology provides uniform control over the false alarm rate across time, i.e.,

$$\mathbb{P}_{H_0} (\exists t \geq 1 : \ \Phi_m(E_1, \ldots, E_t) = 1) \leq \alpha_{\text{source}} + \alpha_{\text{prod}}. \tag{7}$$

It also makes no assumptions about the data distribution or the type of shift. However, the reliance on immediate access to ground truth production labels at each time $t$ limits the method's practical applicability. We now propose a solution that avoids the need for production labels.

# 4 Sequential Harmful Shift Detection without Production Labels

This section consists of two subsections, each detailing one of the two stages of our proposal. The initial stage consists of fitting an error estimator and calibrating it to identify high-error observations with few mistakes. Following this, we apply confidence sequence methods to track the proportion of high errors over time in production, and develop a test for raising an alarm based on this proportion.

## 4.1 Fitting and Calibrating the Error Estimator

The primary drawback of the Podkopaev and Ramdas [2022] method is its reliance on having ground truth labels for the production data, which are often unavailable in real-world scenarios. A straightforward solution is to use a *plug-in* approach: replace the true error in production with an estimated error obtained from a secondary predictive model, denoted as $\hat{r} : \mathcal{X} \to \mathcal{E}$. This model trained to predict the true error of the primary model using any available labeled data. We can then reformulate the alarm function of Equation 6 to deal with unlabeled production data as follows:

$$\hat{\Phi}_m(\boldsymbol{X}_1, \ldots, \boldsymbol{X}_t) = \mathbb{1}\left\{\hat{L}(\hat{r}(\boldsymbol{X}_1), \ldots, \hat{r}(\boldsymbol{X}_t)) > \hat{U}(E_1^0, \ldots, E_n^0) + \epsilon_{\text{tol}}\right\} \tag{8}$$

If $\hat{r}(\cdot)$ is sufficiently accurate, the performance of this alarm mechanism should align closely with what would be achieved if ground truth labels were available. However, even if the estimator $\hat{r}$ exhibits strong performance on its training distribution, the absence of labels in production makes it difficult to conclusively determine the alarm's reliability in a shifting production environment.

Our strategy to address this issue consists of using a calibration step to derive a more reliable statistic from the imperfect estimator $\hat{r}(\cdot)$. Specifically, we propose to track the proportion of observations above a carefully-selected quantile of estimated error, rather than the mean value as in the original method of Podkopaev and Ramdas [2022]. The fundamental hypothesis here is that an estimator, even if not particularly accurate at predicting error magnitudes, may still effectively distinguish between the lowest and highest errors across a dataset, thereby preserving most ordinal relationships between observations. For example, if $\hat{r}(\cdot)$ has correctly represented some underlying patterns to predict the errors, and if $k$-th and $l$-th ranked errors are significantly different, then it is highly probable that $\hat{r}(\boldsymbol{X}_{(k)}) \leq \hat{r}(\boldsymbol{X}_{(l)})$. Focusing on the aggregate distinction of low and high errors rather than the prediction of specific magnitudes allows us to utilize an imperfect estimator $\hat{r}$ more effectively.

Our proposed calibration process is as follows. Given the labeled source data $\mathcal{D}_n$ and a trained error estimator $\hat{r}$, we identify an empirical quantile of the true errors, $q = \mathcal{Q}(p, \{E_i^0\}_{i=1}^n), p \in [0.5, 1)$, and an empirical quantile for the estimated errors $\hat{q} = \mathcal{Q}(\hat{p}, \{\hat{r}(\boldsymbol{X}_i^0) : \boldsymbol{X}_i^0 \in \mathcal{D}_n\}), \hat{p} \in (0, 1)$, such that the selector function $S_{\hat{r},\hat{q}}(\boldsymbol{X}) = \mathbb{1}\{\hat{r}(\boldsymbol{X}) > \hat{q}\}$ reliably distinguishes between observations with true error below and above $q$. Specifically, we seek to balance the statistical power and false discovery proportion (FDP) of the selector, which are defined as follows:

$$\text{Power} = \frac{\sum_{i=1}^n S_{\hat{r},\hat{q}}(\boldsymbol{X}_i^0) \times \mathbb{1}\{E_i^0 > q\}}{\sum_{i=1}^n \mathbb{1}\{E_i^0 > q\}}; \quad \text{FDP} = \frac{\sum_{i=1}^n S_{\hat{r},\hat{q}}(\boldsymbol{X}_i^0) \times \mathbb{1}\{E_i^0 \leq q\}}{\sum_{i=1}^n S_{\hat{r},\hat{q}}(\boldsymbol{X}_i^0)}. \tag{9}$$

We search over a uniform grid of quantile pairs $(p, \hat{p})$, compute the associated thresholds $(q, \hat{q})$, and identify those that achieve an FDP below a maximum value. Among these qualifying pairs, we select the one that maximizes the power. Figure 2 illustrates this process for a toy example. In this case, thresholds are found that achieve a selector power of 0.72 while keeping FDP below the specified maximum of 0.2.

We now present empirical evidence that it is possible to achieve high power and a controlled FDP in realistic settings,

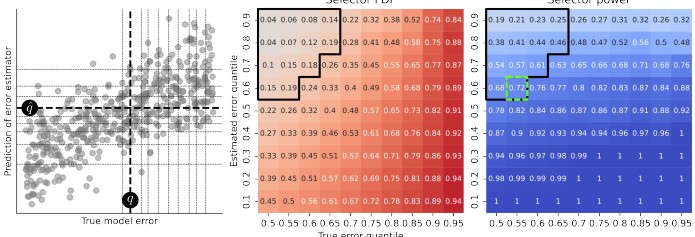

Figure 2: Calibration toy example. **Left:** threshold grid created by sweeping $p \in [0.5, 0.95]$ at increments of 0.05 and $\hat{p} \in [0.1, 0.9]$ at increments of 0.1. **Middle:** FDP of selector for each $(p, \hat{p})$ pair. Black outline indicates pairs for which FDP $< 0.2$. **Right:** selector power for each $(p, \hat{p})$ pair. Green dotted outline indicates the pair that maximises power subject to the FDP $< 0.2$ limit. Corresponding thresholds $(q, \hat{q})$ shown as thick lines in left plot.

using the California house prices [Dua and Graff, 2017], Bike sharing demand [Fanaee-T, 2013], HELOC [FICO, 2018] and Nhanesi [CDC, 1999-2022] datasets. We partition each dataset into training (60%), test (20%) and calibration (20%) sets and use the training data to train random forests (RFs) as the primary models. However, we first ablate the training data in various ways to ensure the models perform poorly on certain subgroups. The ablation is done on a per-feature basis. For continuous features, we exclude 80% of observations with values either above or below the median. For categorical features, we exclude data from one category. We then simulate production environments by gradually reintroducing these previously excluded observations alongside the test set. For each dataset, the number of distribution shifts studied equals the number of features times the number of splits: two for continuous features and the number of categories for discrete ones. We use half of the calibration sets to train RF regressors as the error estimators, then use the remainder to calibrate true and estimated error thresholds using the grid search process described above.

In Figure 3, we present the distribution of the FDP and power across all datasets and shifts, relative to the performance of the error estimator, as measured by the R-squared score on the source/calibration data. The R-squared score is binned into quantiles, with 10 bins used. We observe that the estimators are generally highly imperfect, with R-squared values consistently below 0.3. Despite these low predictive accuracies, we can still find threshold pairs that achieve an FDP below 0.2 in the source data (shown next to the red boxplot). The power ranges from 0.4 for the least accurate estimators to 0.9 for the

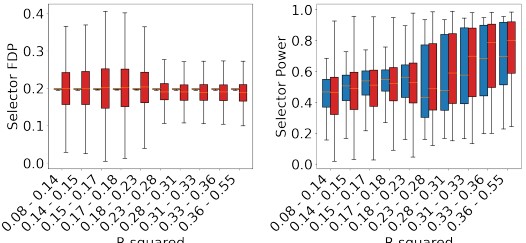

Figure 3: Selector FDP (**left**) and power (**right**) vs estimator accuracy. Results on source data in blue; results on production data in red.

most accurate. Crucially, when we apply the calibrated thresholds in the production environments, we achieve similarly low FDP values (shown in red), almost always below 0.25 (though some reach 0.4), while the power remains similar to the source data, ranging from 0.4 to 0.9. This consistency of the FDP/power even when error estimators are not particularly accurate is promising for shift detection.

## 4.2 Sequential Testing Framework and Performance Guarantees

We can now state the specific objective of our sequential testing framework. During production, we propose to test if there is an increase in the proportion of observations exceeding the true error quantile $q$ obtained in calibration. This is formalized in terms of the following hypotheses:

$$H_0 : \forall t \geq 1, \ \frac{1}{t} \sum_{k=1}^{t} \mathbb{P}_{P^k}(E > q) \leq \mathbb{P}_{P^0}(E > q) + \epsilon_{\text{tol}}, \tag{10}$$

$$H_1 : \exists t \geq T : \frac{1}{t} \sum_{k=1}^{t} \mathbb{P}_{P^k}(E > q) > \mathbb{P}_{P^0}(E > q) + \epsilon_{\text{tol}}, \tag{11}$$

where $\mathbb{P}_{P^k}$ denotes a probability taken under distribution $P^k$. Note that this is a special case of the general test in Equations 1 and 2, with the probability $\theta(P_E^k) = \mathbb{P}_{P^k}(E > q)$ as the error parameter.

Since we cannot observe production errors directly, we use the selector function $S_{\hat{r},\hat{q}}(\boldsymbol{X})$ as a proxy for a check on the true error $E > q$. The effectiveness of the sequential test under this substitution depends on how well the selector's power and FDP properties generalize from the source distribution to the production environment. In particular, we can show that the method outlined below provably controls the false alarm rate given in Equation 3 if the following assumption holds:

**Assumption 4.1.** $\forall t \geq 1, \ \frac{1}{t} \sum_{k=1}^{t} \mathbb{P}_{P^k} \left( S_{\hat{r},\hat{q}}(\boldsymbol{X}) = 1, E \leq q \right) \leq \mathbb{P}_{P^0} \left( S_{\hat{r},\hat{q}}(\boldsymbol{X}) = 1, E \leq q \right).$

Referring back to the example in Figure 2 (left), this assumption implies that at all times during production, the proportion of data observed so far falling the quadrant above and to the left of the calibrated thresholds (---) does not exceed that observed under the source distribution. While we do not claim that this assumption always holds exactly, we find that it is only violated to a small extent in realistic settings (see Appendix A for more discussion and experimental analysis). If this is the case, and thresholds $(q, \hat{q})$ have been found that yield a small number of false discoveries in calibration, $\mathbb{P}_{P^0} \left( S_{\hat{r},\hat{q}}(\boldsymbol{X}) = 1, E \leq q \right)$, then the number of false discoveries in production will also remain low. A substantial increase in false discoveries in production would require a shift specifically targeting those rare observations with low error but high estimated error.

With this foundation established, we can now describe our testing methodology. Following a similar approach to that used by Podkopaev and Ramdas [2022], we construct:

1. A lower bound of $\frac{1}{t} \sum_{k=1}^{t} \mathbb{P}_{P^k}(E > q)$ using a confidence sequence.
2. An upper bound of $\mathbb{P}_{P^0}(E > q)$ using a traditional confidence interval.

To construct the lower bound, we rewrite the target quantity as follows:

$$\frac{1}{t} \sum_{k=1}^{t} \mathbb{P}_{P^k}(E > q) = \frac{1}{t} \sum_{k=1}^{t} \mathbb{P}_{P^k}(S_{\hat{r},\hat{q}}(\boldsymbol{X}) = 1, E > q) + \mathbb{P}_{P^k}(S_{\hat{r},\hat{q}}(\boldsymbol{X}) = 0, E > q) \quad (12)$$

$$\geq \frac{1}{t} \sum_{k=1}^{t} \mathbb{P}_{P^k}(S_{\hat{r},\hat{q}}(\boldsymbol{X}) = 1, E > q) \quad (13)$$

$$= \frac{1}{t} \sum_{k=1}^{t} \mathbb{P}_{P^k}(S_{\hat{r},\hat{q}}(\boldsymbol{X}) = 1) - \frac{1}{t} \sum_{k=1}^{t} \mathbb{P}_{P^k}(S_{\hat{r},\hat{q}}(\boldsymbol{X}) = 1, E \leq q) \quad (14)$$

$$\geq \frac{1}{t} \sum_{k=1}^{t} \mathbb{P}_{P^k}(S_{\hat{r},\hat{q}}(\boldsymbol{X}) = 1) - \mathbb{P}_{P^0}(S_{\hat{r},\hat{q}}(\boldsymbol{X}) = 1, E \leq q). \quad (15)$$

The last inequality uses Assumption 4.1 to substitute the probability of a false discovery in production with the probability on the source. As we can empirically estimate both $\mathbb{P}_{P^k}(S_{\hat{r},\hat{q}}(\boldsymbol{X}) = 1)$ and $\mathbb{P}_{P^0}(S_{\hat{r},\hat{q}}(\boldsymbol{X}) = 1, E \leq q)$ (via the labeled source data $\mathcal{D}_n$), we can use a confidence sequence to construct a valid time-uniform lower bound of their sum. Specifically, we define the bound $\hat{L}_q$ as

$$\hat{L}_q = \frac{1}{t} \sum_{k=1}^{t} \mathbb{1}\left\{S_{\hat{r},\hat{q}}(\boldsymbol{X}_k) = 1\right\} - \frac{1}{n} \sum_{i=1}^{n} \mathbb{1}\left\{S_{\hat{r},\hat{q}}(\boldsymbol{X}_i^0) = 1, E_i^0 \leq q\right\} - w_t - w_n, \quad (16)$$

where $w_t$ and $w_n$ are the widths of the lower and upper bounds of $\frac{1}{t} \sum_{k=1}^{t} \mathbb{P}_{P^t}(S_{\hat{r},\hat{q}}(\boldsymbol{X}) = 1)$ and $\mathbb{P}_{P^0}(S_{\hat{r},\hat{q}}(\boldsymbol{X}) = 1, E \leq q)$ with miscoverage levels $\alpha_1$ and $\alpha_2$ respectively, such that for a total miscoverage level $\alpha_{\text{prod}} = \alpha_1 + \alpha_2 \in (0, 1)$,

$$\mathbb{P}\left(\forall t \geq 1 : \frac{1}{t} \sum_{k=1}^{t} \mathbb{P}_{P^k}(E > q) \geq \hat{L}_q\right) \geq 1 - \alpha_{\text{prod}}. \quad (17)$$

The specific values of $w_t$ and $w_n$ used in our experiments are given in Appendix B. Respectively, these choices correspond to the predictably-mixed empirical-Bernstein (PM-EB) confidence sequence described by Podkopaev and Ramdas [2022], and the classic Hoeffding interval.

We similarly compute an upper bound $\hat{U}_q$ for $\mathbb{P}_{P^0}(E > q)$ as follows:

$$\hat{U}_q = \frac{1}{n} \sum_{i=1}^{n} \mathbb{1}\{E_i^0 > q\} + w_n, \quad (18)$$

where $w_n$ is the same as above. This bound satisfies a miscoverage level $\alpha_{\text{source}} \in (0, 1)$, such that

$$\mathbb{P}\left(\mathbb{P}_{P^0}(E > q) \leq \hat{U}_q\right) \geq 1 - \alpha_{\text{source}}. \quad (19)$$

Finally, we define our sequential test using the following alarm function:

$$\Phi_q(\boldsymbol{X}_1, \ldots, \boldsymbol{X}_t) = \mathbb{1}\left\{\hat{L}_q > \hat{U}_q + \epsilon_{\text{tol}}\right\}, \quad (20)$$

where the subscripted $q$ denotes that we are now detecting shifts in error across a particular quantile, rather than the mean. In Appendix C, we provide a proof of the following statement:

**Theorem 4.2.** *Under Assumption 4.1, $\hat{L}_q$ and $\hat{U}_q$ satisfy Equations 17 and 19. Therefore, the function $\Phi_q$ has false alarm control, i.e.,*

$$\mathbb{P}_{H_0}(\exists t \geq 1 : \Phi_q(X_1, \ldots, X_t) = 1) \leq \alpha_{source} + \alpha_{prod}. \quad (21)$$

While a controlled false alarm rate is a desirable property, the power of $\Phi_q$ may be limited if the degree of error change is not large. Noting that $(1/t) \sum_{k=1}^{t} \mathbb{P}_{P^k}(E > q)$ is lower-bounded by $(1/t) \sum_{k=1}^{t} \mathbb{P}_{P^k}(S_{\hat{r},\hat{q}}(\boldsymbol{X}) = 1, E > q)$, detecting a change requires this probability to exceed $\mathbb{P}_{P^0}(E > q)$. Thus, we also propose to compare $\frac{1}{t} \sum_{k=1}^{t} \mathbb{P}_{P^k}(S_{\hat{r},\hat{q}}(\boldsymbol{X}) = 1, E > q)$ directly with $\mathbb{P}_{P^0}(S_{\hat{r},\hat{q}}(\boldsymbol{X}) = 1, E > q)$. This leads to a second test with higher power. It uses an upper bound of $\mathbb{P}_{P^0}(S_{\hat{r},\hat{q}}(\boldsymbol{X}) = 1, E > q)$, defined as:

$$\hat{U}_q^2 = \frac{1}{n} \sum_{i=1}^{n} \mathbb{1}\{S_{\hat{r},\hat{q}}(\boldsymbol{X}_i^0) = 1, E_i^0 > q\} + w_n, \quad (22)$$

satisfying

$$\mathbb{P}\left(\mathbb{P}_{P^0}(S_{\hat{r},\hat{q}}(\boldsymbol{X}) = 1, E > q) \leq \hat{U}_q^2\right) \geq 1 - \alpha_{\text{source}}. \quad (23)$$

The alarm function for the second test is defined as:

$$\Phi_q^2(\boldsymbol{X}_1, \ldots, \boldsymbol{X}_t) = \mathbb{1}\left\{\hat{L}_q > \hat{U}_q^2 + \epsilon_{\text{tol}}\right\}. \quad (24)$$

Through an almost identical proof, we can similarly show that $\Phi_q^2$ also has false alarm control for comparing $(1/t) \sum_{k=1}^{t} \mathbb{P}_{P^k}(S_{\hat{r},\hat{q}}(\boldsymbol{X}) = 1, E > q)$ with $\mathbb{P}_{P^0}(S_{\hat{r},\hat{q}}(\boldsymbol{X}) = 1, E > q)$.

# 5 Experiments

In this section, we compare the performance of the plug-in approach of Podkopaev and Ramdas [2022]'s method (Equation 8), which is designed to detect a change in the mean error, and our approach, which determines if an increasing number of observations fall beyond a certain quantile. We focus on the second test (Equation 24) to simplify the comparison with the mean detector and because it consistently outperforms the first statistics. Results for the first test are reported in Appendix E. We conduct three experiments using a variety of datasets and setups. The first experiment aims to illustrate the different approaches and demonstrate the applicability of our method to image data and deep learning models. The second experiment returns to the tabular datasets studied in Section 4.1, going into more detail by comparing the mean and quantile detection approaches in terms of power and FDP on the numerous generated shifts. The final experiment also consists of a large-scale evaluation of the approaches, in this case on natural shifts due to temporal and geographical changes. Although the focus of this paper is on the sequential or online setting, we provide an analysis using state-of-the-art methods in the batch setting in Appendix F.

## 5.1 Illustrative Example on an Image Dataset

The first experiment replicates the setup of Saerens et al. [2002] using the CelebA dataset [Liu et al., 2015]. They demonstrate that a ResNet50 model [He et al., 2016] trained on this dataset performs poorly on "males with blond hair" due to spurious correlations. We split this dataset into a training set (60%), test set (20%) and calibration set (20%), and train a ResNet50 on the training set. Using half of the calibration set, we train another ResNet50 (with a regression head) as an error estimator. The remaining half is employed to determine the empirical quantiles $p \in [0.5, 1), \hat{p} \in (0, 1)$ at which we achieve maximum power while keeping the FDP below 0.2. We create a harmful shift in production as follows. For each time step up to $t = 4990$, we sample an observation uniform-randomly from the test set. Thereafter, we begin to oversample instances of males with blond hair, sampling such an observation with probability $\beta_t = 1/(1 + \exp(-(t - 4990)))$, and a random observation otherwise.

The objective of this experiment is to visually observe how the methods can be used to monitor performance shift over time and to evaluate how each method compares to an idealised version with access to true production errors. Both Podkopaev and Ramdas [2022]'s method (mean detector) and our approach (quantile detector) involve comparing a lower bound to an upper bound. For both methods, Figure 4 displays the lower bound in blue and the version calculated with true production errors in gray. For the quantile detector, the blue line corresponds to $\hat{L}_q$ of Equation 16, which is the estimated lower bound of $\frac{1}{t}\sum_{k=1}^{t} \mathbb{P}_{P^k}(E > q)$ with estimated production errors. The gray line represents the lower bound of this same quantity, except computed using the true errors. The blue lower bound of the plug-in approach of the mean detector is defined as $\hat{L}(\hat{r}(\boldsymbol{X}_1), \ldots, \hat{r}(\boldsymbol{X}_t))$. The gray line represents $\hat{L}(E_1, \ldots, E_t)$, the lower bound of the original mean detector using true errors. The upper bound that needs to be surpassed for each method to raise an alarm is depicted in red. For the quantile detector, this is the second lower bound $\hat{L}_q^2$, and for the mean detector, it is $\hat{U}(\hat{r}(\boldsymbol{X}_1), \ldots, \hat{r}(\boldsymbol{X}_t))$. For the quantile detector, we also plot in pink the upper bound of the first statistic $\Phi_q$ (Equation 20).

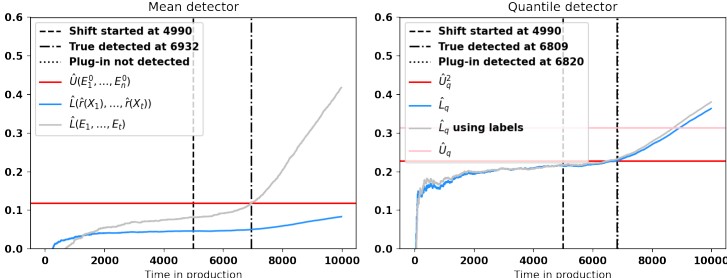

Figure 4: Evolution of bounds in production for mean detector **(left)** and quantile detector **(right).**

The R-squared value of the error estimator on the source distribution is 0.35, which is not especially high. By analyzing the upper bounds for each method in Figure 4, we observe that all bounds remain roughly constant before the shift starts. Unsurprisingly, we observe the mean detector using true

production errors quickly detects the shift (gray line). In contrast, its plug-in version raises an alarm with a significantly delayed detection. For the quantile detector, there is a much smaller difference between the lower bound of the plug-in and the one using true production errors. This observation validates our expectation that the FDP remains relatively stable post-shift. Additionally, as expected, the plot shows that the lower bound of the quantile detector crosses the upper bound of the second statistic (red line) much earlier than that of the first statistic (pink line).

This experiment suggests that in scenarios with a less accurate error estimator, targeting quantile changes is more effective for detecting harmful shifts than focusing on mean change. Additional experiments on image datasets confirming this observation can be found in Appendix D. Larger-scale analyses in the following subsections examine the advantages of the quantile detector in more depth.

## 5.2 Synthetic Shifts on Tabular Datasets

In this section, we conduct a large-scale experiment to evaluate the effectiveness of both methods in detecting harmful shifts while maintaining their ability to control false alarms. We also analyze how these metrics relate to the performance of the error estimator. We use two regression datasets (California house prices and bike sharing demand) as well as two classification datasets (HELOC and Nhanesi). We follow the feature-splitting setup of Section 4.1 to generate synthetic distribution shifts, excluding splits that result in subsets with fewer than 10 observations, and repeat each split 50 times with different random seeds.

Table 1 shows the number of generated shifts and the number of harmful shifts detected by each method using the true errors (H-M for mean detector and H-Q for quantile detector). A shift is considered harmful by each method as soon as the lower bound exceeds the upper bound plus $\epsilon_{tol} = 0$.

Table 1: Description of the shifts generated.

| Data | # Generated Shifts | H-M | H-Q |
|---|---|---|---|
| california | 62 | 10 | 48 |
| bike | 2129 | 57 | 961 |
| heloc | 3385 | 774 | 1283 |
| nhanesi | 1697 | 377 | 679 |

The left plot of Figure 5 displays the aggregated results across all distribution shifts for mean detection (red) and quantile detection (green) on the different datasets. The points labeled "all-[method]" represent the average results across the datasets. The quantile method achieves a significantly better power-FDP balance: (power 0.83, FDP 0.11) compared to the mean method: (power 0.67, FDP 0.41) across all experiments. An exception is observed for the Nhanesi dataset, where the mean detection shows slightly better power. However, overall, the quantile detection demonstrates a superior trade-off between power and false alarms. A similar trend is observed in the middle plot, which analyzes the absolute difference in detection time between each method using estimated errors and the same method with access to true errors. In the right plot, we compute how the power across datasets varies when we increase the threshold at which we consider the true shift as harmful ($\epsilon_{tol}$). Across varying intensities of shift, the quantile detector consistently outperforms the mean detector, with false alarm rates at $\epsilon_{tol} = 0$ being 0.41 and 0.11, respectively.

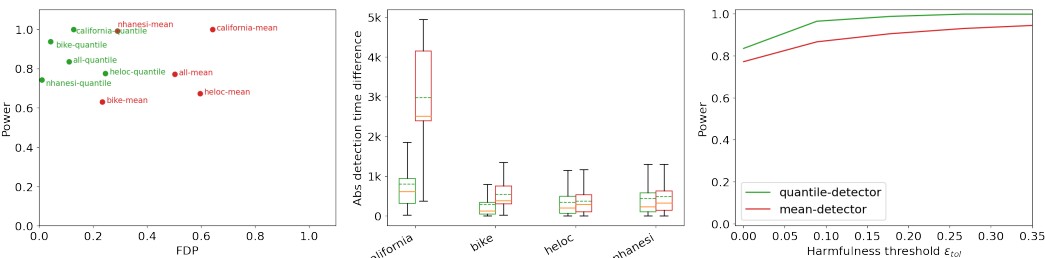

Figure 5: **Left:** Power/FDP when $\epsilon_{tol} = 0$ for all datasets. **Middle:** Absolute detection time difference vs. the methods using true errors. **Right:** Power values for different harmfulness thresholds ($\epsilon_{tol}$).

In Figure 6, we further investigate the relationship between the power (top row) and FDP (bottom row) of each method and the error estimator's performance binned into 10 quantiles for each dataset. The error estimator performance, measured by R-squared values, is generally low across all experiments (0.10 - 0.26). Notably, the quantile detector consistently maintains a lower FDP compared to the

mean detector across all error estimator values. Regarding power, excluding the Nhanesi dataset, the quantile detector performs better than or equal to the mean detector.

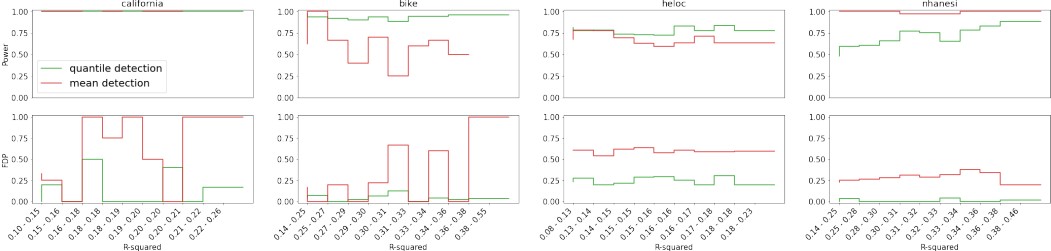

Figure 6: Power and FDP by error across all datasets.

## 5.3 Natural Shifts on a Tabular Dataset

In the last experiment, we conduct another large-scale evaluation of our approach on natural shifts within the Folktables dataset [Ding et al., 2021]. This dataset is derived from US Census data spanning all fifty states within the US (plus Puerto Rico), each with a unique data distribution. Furthermore, it includes data from multiple years (from 2014 to 2018), introducing a form of temporal distribution shift in addition to the variations between states. We select the income feature as the target label, specifically predicting whether income exceeds $50,000$. We first split the dataset of each state in the year 2014 into training ($50\%$), and calibration ($50\%$). Then, we train a separate RF classifier in each state in the year 2014, and an RF regressor to learn the error of the primary model on the calibration set. Subsequently, we evaluate the model's error on all the remaining 50 states over 5 years, effectively creating 250 production datasets. We consider a shift to be harmful if the model's error in production exceeds the error on the calibration dataset plus $\epsilon_{tol} = 0$. We introduce the shift in all datasets starting at time $t = 3300$.

Table 2 summarizes the results for both methods, demonstrating that the quantile detector consistently outperforms the mean detector across all metrics.

Table 2: Comparison of detection methods on Folktables data.

| Method | Power | FDP | Mean detection time |
|---|---|---|---|
| Quantile detector | 0.48 | 0.019 | 3727 |
| Mean detector | 0.01 | 0.19 | 4945 |

Figure 7 plots the sensitivity of each method relative to the shift harmfulness threshold. We observe that the quantile detector maintains superior performance across all threshold values.

Overall, this experiment provides good evidence that our proposed method is effective under the kinds of natural shift encountered in realistic production environments.

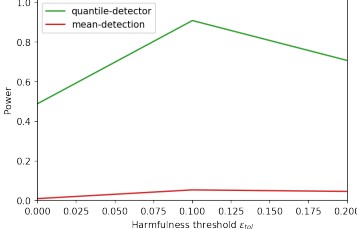

Figure 7: Power corresponding to different levels of the harmfulness threshold ($\epsilon_{tol}$) on Folktables.

## 6 Conclusion

We have introduced an approach to identifying harmful distribution shifts in continuous production environments where ground truth labels are unavailable. Utilizing a plug-in strategy that substitutes true errors with estimated errors, alongside a threshold calibration step, our method effectively controls false alarms without relying on perfect error predictions. Experiments on real-world datasets demonstrate that our approach is effective in terms of detection power, false alarm control and detection time across various shifts, including covariate, label, and temporal shifts. In future work, we plan to apply interpretability techniques to the quantile detector to understand where and how the data are shifting in the input space, and to use this information to improve the primary model itself.

## Disclaimer

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

# A  Discussion of Assumption 4.1

To formalize the statement in the main body of the paper, we do not expect Assumption 4.1 to hold exactly, but we expect that in realistic settings, for all $t \geq 1$, the inequality

$$\frac{1}{t} \sum_{k=1}^{t} \mathbb{P}_{P^k} \left( S_{\hat{r}, \hat{q}}(\boldsymbol{X}) = 1, E \leq q \right) \leq \mathbb{P}_{P^0} \left( S_{\hat{r}, \hat{q}}(\boldsymbol{X}) = 1, E \leq q \right) + \delta_{tol}$$

holds with a small $\delta_{tol}$. For instance, in Figure 8, we compute the empirical distribution estimate of $\delta = \frac{1}{t} \sum_{k=1}^{t} \mathbb{P}_{P^k} \left( S_{\hat{r}, \hat{q}}(\boldsymbol{X}) = 1, E \leq q \right) - \mathbb{P}_{P^0} \left( S_{\hat{r}, \hat{q}}(\boldsymbol{X}) = 1, E \leq q \right)$ with $t$ equals to the total number of production data across the different distribution shifts and datasets of Section 5.2 and the natural distribution shifts of Section 5.3.

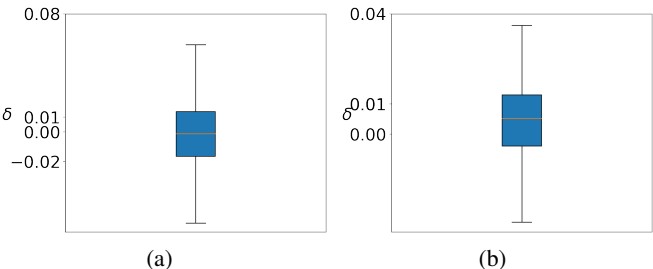

Figure 8: Distribution of $\delta$ across the different shifts and datasets of Section 5.2 (a) and the natural distribution shifts of Section 5.3 (b)

We observe that Assumption 4.1 is valid approximately 50% of the time for both experimental setups, corresponding to the cases where $\delta$ is negative. In the other half of the cases where the assumption is not verified, we note that $\delta$ is very small, often less than 0.01.

It should be noted that Assumption 4.1 allows for controlling false alarms when $\delta$ is zero or negative. To control false alarms when $\delta$ is positive, it is sufficient to always add $\delta$ to the lower bound $\hat{L}_q$ (Eq. 16) to have the false alarm guarantee. Specifically, under the assumption that for all $t \geq 1$, $\frac{1}{t} \sum_{k=1}^{t} \mathbb{P}_{P^k} \left( S_{\hat{r}, \hat{q}}(\boldsymbol{X}) = 1, E \leq q \right) \leq \mathbb{P}_{P^0} \left( S_{\hat{r}, \hat{q}}(\boldsymbol{X}) = 1, E \leq q \right) + \delta$ we can show, similar to Theorem 4.2, that the corrected bounds $\hat{L}_q^{corr} = \hat{L}_q - \delta$ used in the following statistic:

$$\Phi_q^{corr}(\boldsymbol{X}_1, \ldots, \boldsymbol{X}_t) = \mathbb{1} \left\{ \hat{L}_q^{corr} > \hat{U}_q + \epsilon_{\text{tol}} \right\}, \tag{25}$$

will have false alarm control. The proof is identical to the proof of the Theorem 4.2, with $\hat{L}_q$ replaced by $\hat{L}_q^{corr}$.

However, in practice, we do not know the value of $\delta$. Fortunately, in most realistic cases we have observed, $\delta$ is very small, especially compared to our maximum false alarm threshold of 0.2. Therefore, not adding this correction has very little impact on the statistics without correction (Equation 20).

# B   Bounds of the Confidence Sequences and Intervals Used

In our experiments, the confidence sequence bound $w_t$ is that of the Empirical Bernstein confidence sequence, as defined in the Theorem below. For a more detailed presentation of different confidence sequences, we refer the reader to Howard et al. [2021].

**Theorem B.1.** *Let $\hat{\mu}_t = \frac{1}{t} \sum_{i=1}^{t} X_i$, and suppose $X_i$ are bounded a.s. for each $i \geq 1$. Then, for each $\alpha \in (0,1)$,*

$$C_t = \{\theta_t \pm w_t\} \quad \text{forms a } (1-\alpha)\text{-level confidence sequence for } \mathbb{E}(\hat{\mu}_t),$$

*where $w_t = c_\alpha \frac{\sqrt{\hat{V}_t \log\log \hat{V}_t}}{t}$, $\hat{V}_t = \sum_{i=1}^{t} (X_i - \hat{\mu}_{i-1})^2$ denotes an empirical variance term and $c_\alpha \asymp \sqrt{\log(1/\alpha)}$.*

When we use a confidence interval, we use the classic Hoeffding interval:

$$C_n = \{\hat{\mu}_n \pm w_n\} \quad \text{forms a } (1-\alpha)\text{-level confidence interval for } \mathbb{E}(\hat{\mu}_n),$$

where $w_n = \frac{\log(2/\alpha)}{2n}$.

# C   Proofs

**Theorem C.1.** *Under Assumption 4.1, $\hat{L}_q$ and $\hat{U}_q$ satisfy Equations 17 and 19. Therefore, the function $\Phi_q$ has false alarm control, i.e.,*

$$\mathbb{P}_{H_0}\left(\exists t \geq 1: \; \Phi_q(X_1, \ldots, X_t) = 1\right) \leq \alpha_{source} + \alpha_{prod}. \tag{26}$$

*Proof.*

$$\mathbb{P}_{H_0}\{\exists t \geq 1: \; \Phi_q(X_1, \ldots, X_t) = 1\}$$

$$= \mathbb{P}_{H_0}\left\{\exists t \geq 1: \; \hat{L}_q > \hat{U}_q + \epsilon_{\text{tol}}\right\}$$

$$= \mathbb{P}_{H_0}\left\{\exists t \geq 1: \; \left(\hat{L}_q - (1/t)\sum_{k=1}^{t}\mathbb{P}_{P^k}(S_{\hat{r},\hat{q}}(\boldsymbol{X}) = 1)\right) - \left(\hat{U}_q - \mathbb{P}_{P^0}(S_{\hat{r},\hat{q}}(\boldsymbol{X}) = 1)\right)\right.$$

$$\left. > \epsilon_{\text{tol}} - \left((1/t)\sum_{k=1}^{t}\mathbb{P}_{P^k}(S_{\hat{r},\hat{q}}(\boldsymbol{X}) = 1) - \mathbb{P}_{P^0}(S_{\hat{r},\hat{q}}(\boldsymbol{X}) = 1)\right)\right\}$$

$$\leq \mathbb{P}_{H_0}\left\{\exists t \geq 1: \; \left(\hat{L}_q - (1/t)\sum_{k=1}^{t}\mathbb{P}_{P^k}(S_{\hat{r},\hat{q}}(\boldsymbol{X}) = 1)\right) - \left(\hat{U}_q - \mathbb{P}_{P^0}(S_{\hat{r},\hat{q}}(\boldsymbol{X}) = 1)\right) > 0\right\}$$

$$\leq \mathbb{P}_{H_0}\left\{\exists t \geq 1: \; \left(\hat{L}_q - (1/t)\sum_{k=1}^{t}\mathbb{P}_{P^k}(S_{\hat{r},\hat{q}}(\boldsymbol{X}) = 1)\right) > 0\right\} + \mathbb{P}_{H_0}\left\{\left(\hat{U}_q - \mathbb{P}_{P^0}(S_{\hat{r},\hat{q}}(\boldsymbol{X}) = 1)\right) > 0\right\}$$

$$\leq \alpha_{\text{source}} + \alpha_{\text{prod}}$$

The last inequality is due to Equation 17 and 19.   $\square$

# D Additional Experiments on Image Datasets

Here, we conduct two experiments using image datasets, specifically CIFAR-10 [Krizhevsky et al.] and Fashion MNIST [Xiao et al., 2017]. Similar to previous experiments, we remove some part of the data during training phase, here 90% of the observations with label 3 for both datasets, and reintroduce them gradually during the production phase.

In Figure 9, we observe the same behavior as in Section 5.1. The quantile detector detects changes more quickly than the mean detector, and the performance of the former is closer to the true version than that of the latter.

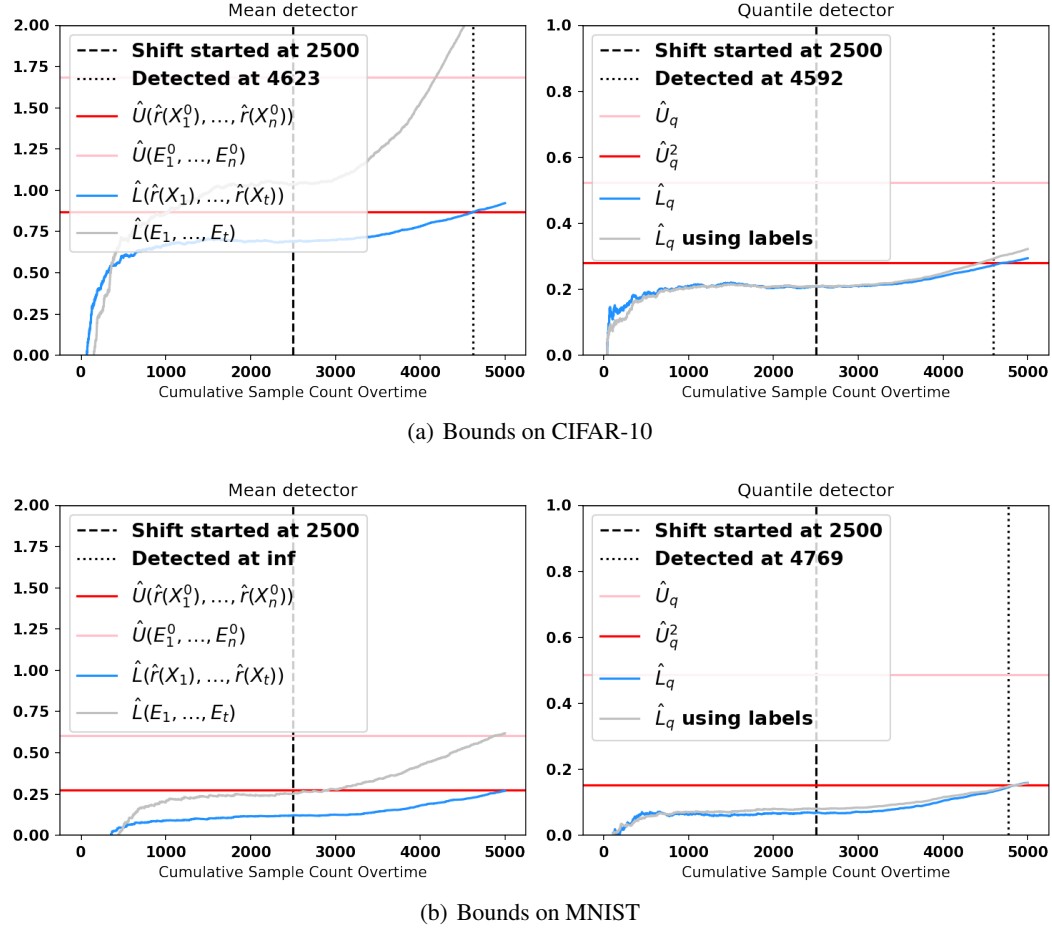

(a) Bounds on CIFAR-10

(b) Bounds on MNIST

Figure 9: Evolution of the bounds in production for mean detector (**left**) and quantile detector (**right**).

# E  Comparison Between $\Phi_q$ and $\Phi_q^2$

In this section, we will revisit the main experiments from sections 5.2 and 5.3, incorporating comparisons with the quantile detector using the first statistic $\Phi_q$. As expected, in figure 10, we consistently observe that the first statistic $\Phi_q$ achieves a better FDP than the second statistic $\Phi_q^2$ at the cost of a much smaller power. In addition, $\Phi_q$ fails to detect any shift in the California dataset and has a much higher detection time.

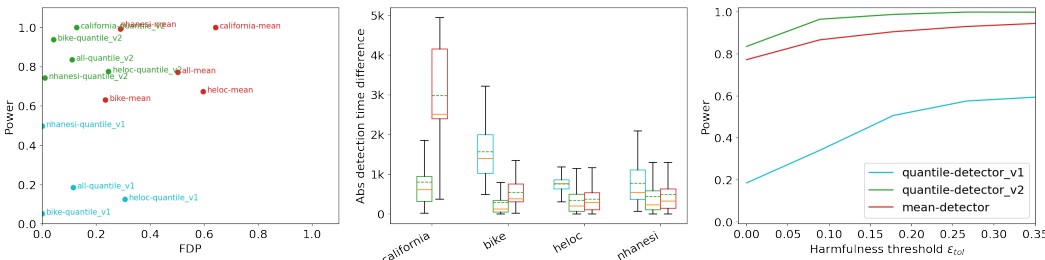

Figure 10: **Left:** Power/FDP when $\epsilon_{tol} = 0$ for all datasets. **Middle:** Absolute detection time difference vs. the methods using true errors. **Right:** Power values for different harmfulness thresholds ($\epsilon_{tol}$).

In Figure 11, we have also computed the power relative to the harmfulness threshold $\epsilon_{tol}$ of the Folktables data from Section 5.3. The second statistic performs much better than the first in terms of power, although the FDP of the latter is smaller ($0.004$) compared to $0.019$ for the former.

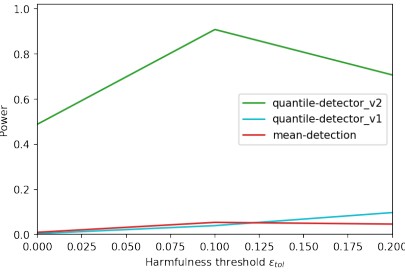

Figure 11: Power corresponding to different levels of the harmfulness threshold ($\epsilon_{tol}$) on Folktables.

# F  Evaluations in Batch Setting

In this section, we compare our approach with a leading method, *Detectron*, proposed by Ginsberg et al. [2022], in a batch setting. Directly comparing our method (SHSD) to those in Ginsberg et al. [2022] presents challenges due to fundamental differences in their design. Their methods are tailored for an offline batch setup, which requires a complete batch of production data to compute statistics and trigger alarms. In contrast, our approach is optimized for an online setting where shifts may occur gradually and continuously, necessitating real-time decisions without the ability to observe an entire unlabelled batch upfront. Our methodology is designed to detect harmful performance shifts on the fly, processing each observation sequentially without requiring access to the full production dataset.

Applying offline methods like Detectron in an online setting would be both impractical and unfair, as these methods rely on training a model or computing statistics from a batch of data. Additionally, it would be computationally expensive since Detectron requires training a new model for each batch of production data. Consequently, deploying this approach online would entail training a number of models proportional to the production data size.

To provide a meaningful comparison, we evaluated our method alongside Detectron in a batch setting, progressively increasing the size of the production or out-of-distribution (OOD) data. We generated shifts in line with the setup described in Section 5.2, ensuring no shift within the first 1300 samples

of production data. We utilized the NHANESI classification dataset, as Detectron is specifically designed for classification tasks. Our experiments were replicated 50 times, yielding a total of 10,200 shift instances.

Table 3: Comparison of Power and FDP metrics for Detectron and SHSD across different OOD sizes.

| OOD Size | Power Detectron | FDP Detectron | Power SHSD | FDP SHSD |
|---|---|---|---|---|
| 100 | N/A | 1.00 | N/A | 0.00 |
| 1000 | N/A | 1.00 | N/A | 0.00 |
| 2000 | 0.96 | 0.61 | 0.40 | 0.02 |
| 3000 | 0.98 | 0.60 | 0.63 | 0.02 |
| 3500 | 0.98 | 0.60 | 0.67 | 0.02 |
| 8593 | 0.98 | 0.60 | 0.74 | 0.04 |

The results, summarized in Table 3, demonstrate that for smaller sample sizes (100 and 1000), our method did not detect any shifts, as expected given the lack of shifts in the initial 1300 samples. However, Detectron raised a significant number of alarms (1126 out of 1700 for sample size 100 and 1493 for sample size 1000), all of which were false alarms. For larger sample sizes, while Detectron shows high power in detecting shifts, it also produces a high false discovery proportion (FDP). In contrast, our method exhibits lower power but significantly better control over false alarms, consistent with our objective of minimizing false positives.

These results confirm that our method performs robustly in both batch and online settings, effectively maintaining low false alarm rates while detecting harmful shifts as they arise.

## F.1 Limitations of Disagreement-based Detectors

In this section, we highlight some potential limitations of disagreement-based detectors, such as Detectron, which may limit their effectiveness in certain contexts.

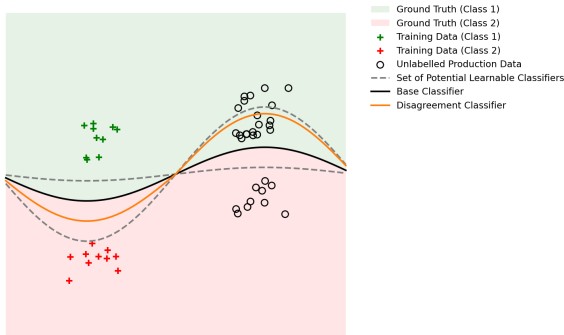

Figure 12: Illustration of a Disagreement-Based Detector Failure Case

The primary concept behind Detectron is to train a disagreement classifier that performs comparably to the original model on the training distribution while disagreeing with the original model's predictions on production data. This approach is highly sensitive to the base model's performance, the choice of function class, and the size and nature of the production data. Although Detectron shows high power in detecting harmful shifts (as evidenced by our experiments), it may raise false alarms when the shift is benign.

In Figure 12, we illustrate a failure case for the disagreement-based detector. In this example, training data points are represented in red and green, with the ground truth shaded accordingly. The solid black line represents the decision boundary of a base model, which we assume to be a perfect classifier. The data has shifted to the right, resulting in unlabeled production data that is still correctly classified by the base model.

We've also depicted the potential learnable classifier as a dashed line, representing the boundary of all possible functions, which depends on the model type, complexity used for the disagreement classifier, and the nature and size of the data. We have shown a potential disagreement classifier in orange that performs similarly to the original model on training data but disagrees with the predictions of the base classifier in the production data. As shown, even with a benign shift, we can still find a disagreement classifier that performs well on training data but disagrees significantly in production, raising a false alarm.

## G    Experimental Compute Resources

We run all our experiments on an Amazon EC2 instance (c5.4xlarge) that consists of 16 vCPUs and 32 GB of RAM.

## H    Impact Statement

This research, focusing on developing algorithms to detect harmful distribution shifts in machine learning models, has significant and diverse practical impacts. It offers a solution to a key challenge in the safe deployment of AI across various industries by detecting shifts without needing labeled data. For instance, in healthcare, the ability to identify harmful shifts in predictive models enhances the accuracy and reliability of diagnostic tools, which is especially vital as patient data continuously changes due to new diseases or demographic shifts. In finance, the algorithms can detect market trends or consumer behaviour changes that might negatively impact forecasting models, leading to more adaptive and resilient economic models, improved risk management, and better-informed decision-making processes.

