# OpenReview forum: "Sequential Harmful Shift Detection Without Labels"
_NeurIPS.cc/2024/Conference — NeurIPS 2024 poster_

### Official Review · Reviewer_MiPF · 2024-06-13

**Soundness:** 2
**Presentation:** 2
**Contribution:** 2
**Rating:** 5
**Confidence:** 1

**Summary:**

The authors introduce an approach for detecting distribution shifts that negatively impact the performance of machine learning models in continuous production environments, which requires no access to ground truth data labels. They solution substitute true errors with the predictions of a learnt error estimator.

**Strengths:**

This work is the first to propose a principled method for detecting harmful distribution shifts without requiring true label.

**Weaknesses:**

It is worth noting that while I am familiar with distribution shifts, I do not know  continuous production environments is limited. I will be evaluating this paper from the perspective of an outsider in this field, and will rely more on the feedback of other reviewers.
**************************************
I encourage the AC to bring in reviewers who are familiar with this field.

I encourage the AC to give less weight to my review comments.
**************************************
Weaknesses:

1) I suggest that the authors provide a section on related work concerning distribution shifts and continuous production environments, to help readers clearly understand the motivation behind this paper. To my knowledge, the Out-of-Distribution (OOD) field’s research on Dataset Shift [1-4] also discusses distribution shifts; please include a discussion on this. The introduction does not well introduce the research questions of this paper, especially considering that this paper has not yet reached the length limit.

2) My major concern about this paper is whether “requires no access to ground truth data labels” is a worthwhile background setting to study, especially since Podkopaev and Ramdas [2022] have already satisfactorily addressed more common scenarios. Considering that the method proposed by the authors is relatively straightforward: training an error estimation model to predict the performance of the primary model on production data.

[1] Puli et al., Don’t Blame Dataset Shift! Shortcut Learning Due to Gradients and Cross Entropy, in NeurIPS 2023.

[2] Huang et al., Harnessing Out-of-Distribution Examples via Augmenting Content and Style, in ICLR 2022.

[3] Silva et al., The Value of Out-of-Distribution Data, in ICML 2023.

[4] Yang et al., Not All Out-of-Distribution Data Are Harmful to Open-Set Active Learning, in NeurIPS 2023.

**Questions:**

N/A

---

> ### Author Rebuttal · Authors · 2024-08-04
>
> We thank the reviewer for taking the time to review our paper.
>
> We appreciate the suggestion to include a dedicated section on related work concerning distribution shifts and continuous production environments. Regarding the question of whether "requires no access to ground truth data labels" is a worthwhile background setting to study, we believe it is highly relevant. In many real-world environments, ground truth labels are either unavailable, delayed, or expensive to obtain. Examples include:
>
> * **Healthcare:** (Raravian et al., 2015, Population-level prediction of type 2 diabetes from claims data and analysis of risk factors)
> * **Insurance:** (Noodhun et al., 2018, Risk prediction in the life insurance industry using supervised learning algorithms)
> * **Predicting Future Outcomes:** (Zhang et al., 2019, Time-aware adversarial networks for adapting disease progression modeling)
> * **High Collection Costs:** (Lieu et al., 2022, Deep unsupervised domain adaptation: A review of recent advances and perspectives)
>
> Regarding the remarks about our method, it is not solely about learning an error estimator. The cornerstone of our method is the calibration step that leverages even an imperfect estimator to achieve performance comparable to methods that have access to ground truth labels. This is particularly useful since it is often unrealistic to expect a highly accurate error estimator in many practical scenarios.

---

> > ### Comment · Reviewer_MiPF · 2024-08-13
> >
> > I have gone through the responses.
> >
> > **I encourage AC to completely ignore my comments and score.**
> >
> > Good luck :)

---

### Official Review · Reviewer_FDg9 · 2024-07-05

**Soundness:** 3
**Presentation:** 4
**Contribution:** 3
**Rating:** 6
**Confidence:** 3

**Summary:**

The paper introduces a sequential drift detector designed to identify drifts that may negatively impact model performance without requiring target labels in production. This is achieved by training an error proxy model on a calibration dataset, which can then be applied in a production scenario, thereby eliminating the need for target labels. While the practical benefits of this method are evident, its performance is initially worse than the reference method that uses labels. This issue is addressed by proposing a modification to the scoring metric, making the performance of both methods comparable. Although the method that uses labels detects drifts earlier, the label-less method now demonstrates similar performance. The methods are tested and compared on multiple synthetic and real-world datasets.

**Strengths:**

### Originality

- **Novel Approach:** The introduction of a sequential drift detector that does not require target labels in production is a welcomed addiotion in the field of drift detection.
- **Error Proxy Model:** The use of an error proxy model trained on a calibration dataset to identify drifts is a powerful contribution that addresses the challenge of label scarcity in production environments. This approach also opens up the possibility to use specialized error proxy models in the future to extend the approach.

### Quality

- **Thorough Evaluation:** The methods are rigorously tested and compared on multiple synthetic and real-world datasets, providing a comprehensive evaluation of the proposed approach.

### Clarity

- **Clear Explanation:** The paper clearly explains the methodology, including the training of the error proxy model and its application in production scenarios.
- **Well designed mathematical Notation**: The provided notation is intuitive and provides detailed introspection into the newly proposed parts of the method.

### Significance

- **Practical Relevance:** The method's ability to function without target labels in production is highly relevant for real-world applications where obtaining labeled data is often challenging.
- **Comparable Performance:** By achieving performance comparable to traditional methods that use labels, the proposed method offers a valuable alternative for drift detection in label-scarce environments.

**Weaknesses:**

### Line 17-19:

- Domain knowledge regarding performance-based drift detection methods is not mentioned.
- The claims regarding the properties of “traditional” drift detection methods are unfounded and can be seen as misinformation.
- For further references the survey "Learning under Concept Drift: A Review” of  Jie Lu, Fellow, IEEE, Anjin Liu, Member, IEEE, Fan Dong, Feng Gu, Jo˜ao Gama, and Guangquan Zhang provides a structured overview of different detection mechanisms and differences.
- In addition, the use of a reference window is presented as a weakness, but the proposed method also uses a calibration set, which can be seen as a reference window.

### Line 134:

- Phrases like “If sufficient”, “Should align”, “would achieve” involve a lot of assumptions.
- If the error estimator is a statistical model, it would open this expression up to further introspection.
- While the statements proposed by the work seem solid, sections where these subjective claims are mentioned (e.g., here and the beginning of section 2) are very loose and are left unanswered.

### Line 193:

- The FDP plot has no blue in it; what does this sentence refer to?

### Equations 10 and 11:

- These equations confuse more than they clarify.
- A new form is introduced just to show how it can be directly inferred from Equations 1 and 2. This, by itself, would be fine, but then the notation is changed again directly.
- Why not go from Equations 1 and 2 directly to Assumption 4.1?

### Experimental Reproducibility:

- While the data is well explained, the configuration of the experiments is left up to interpretation.
- Access to the scripts used to run the experiments would highly benefit reproducibility.
- While the paper checklist mentions that the code will be provided, this is currently not realized, making it impossible to include it in the review.

**Questions:**

### Line 7:

- Is "learnt" the old variation of "learned" or is there a specialized meaning to it?

### Line 46:

- Does the predictor for the model error serve as an approximation of the conditional probability to produce a higher error in areas of the feature space with sparse or contradictory behavior?
- Would the representation of this not be best captured by a Gaussian Process model?
- You do not mention the model you used to produce your results. (Or do you always use the same model for the main prediction and the error prediction, if yes, why?)

### Line 74:

- In which domain do X and Y reside?
- What are we trying to predict given what kind of information?
- The current specification does not allow the assumption of a suitable bounded error function. Specifying this kind of error for regression problems proves to be very difficult and highly dependent on the application context, for example.

### General Question on DDM and PHT:

- The DDM (Gama, J., Medas, P., Castillo, G., Rodrigues, P. (2004). Learning with Drift Detection. In: Bazzan, A.L.C., Labidi, S. (eds) Advances in Artificial Intelligence – SBIA 2004. SBIA 2004.) and PHT (Page, E.S.: Continuous inspection schemes. Biometrika 41(1-2), 100–115 (1954). DOI 10.1093/biomet/41.1-2.100. and Gama, J., Sebastião, R. & Rodrigues, P.P. On evaluating stream learning algorithms. Mach Learn 90, 317–346 (2013). https://doi.org/10.1007/s10994-012-5320-9) drift detection tests are frequently cited and also track the development of the mean model error over time and restrict its allowed deviation based on sequential analysis. How does this compare to your work, and what are the differences?

### Line 186:

- Which type of R² score is used here? What is the reference model?
- How is the R² score used in the production scenario, as it is batched based the application on sequential data points is not natively defined.
- Against your assumption in section 2, the R² score is not bounded! Depending on the formulation, it can have large positive values but be bounded to larger than 0 or have large negative values but be always smaller than 1. R² is only bounded if it is viewed as a data descriptor that assumes that a linear model is used to describe the data, then it is a measure for the explainable variance. This specification confuses more than it help. The R²  formulation you use should be clearly specified in mathematical notation.

### Equations 12 to 15:

- Are you referring to joint probabilities or conditional probabilities? So, S(X)=1 and E>q or S(X)=1 given E>q?

### Line 235:

- If the Hoeffding interval is based on the Hoeffding Inequality, does your application fulfill all stated requirements?
- With special focus on the R² Score you use as your “bounded” metric, Theorem B.1 seems to have unfulfilled assumptions.

### Section 5.1:

- What error metric is used for the stated experiment? Accuracy?

**Limitations:**

The limitations are mentioned and discussed.

The assumptions used for some proves are however not discussed into detail. Further Details in Questions and Weaknesses

---

> ### Author Rebuttal · Authors · 2024-08-04
>
> We thank the reviewer for taking the time to review our paper and acknowledging our contributions.
>
> > The claims regarding the properties of “traditional” drift detection methods are unfounded and can be seen as misinformation.
>
> Regarding the issue raised about traditional shift methods (Lines 17-19), specifically their inability to distinguish harmful from benign shifts and their inflated false alarm rates, our assertions are based on the findings of Podkopaev and Ramdas (2022). Their analysis in Appendix A ("Issues with existing tests for distribution shifts/drifts") of their paper supports our claims. We will cite this appendix directly in the final version of our paper.
>
> > Phrases like “If sufficient”, “Should align”, “would achieve” involve a lot of assumptions.
>
> Yes, we agree. In Line 134, we were describing the naive approach of using a plug-in error estimator. This served as a prelude to introducing our strategy, which aims to achieve good performance using an imperfect proxy for model performance.
>
> > If the error estimator is a statistical model, it would open this expression up to further introspection.
>
> We agree. Additionally, as the reviewer highlighted, our approach allows for the use of any proxy, not only error estimation. For instance, in classification problems, we can use calibrated probabilities instead of the prediction of the error estimator as a proxy before our calibration step. We have tested this approach and found that it demonstrated similar performance to the error estimator.
>
> > The FDP plot has no blue in it; what does this sentence refer to?
>
> The variance of the distribution was so small that it appeared as a line, making the blue colour not visible as in the production data case. We will clarify this in the final version.
>
> > Why not go from Equations 1 and 2 directly to Assumption 4.1?
>
> We will simplify the presentation by going directly from Equations 1 and 2 to Assumption 4.1.
>
> > Is "learnt" the old variation of "learned" or is there a specialized meaning to it?
>
> Yes, it is the British variant, and they both have the same meaning.
>
> > Does the predictor for the model error serve as an approximation of the conditional probability to produce a higher error in areas of the feature space with sparse or contradictory behavior? Would the representation of this not be best captured by a Gaussian Process model?
>
> Yes, exactly. It is an implicit way of defining the regions where the model performs poorly. Particularly in the case of tabular data, we have considered tree-based models as they are well-known to work best. In addition, it opens the possibility of examining the leaves of the tree to define hyper-rectangular region. These regions can be represented as rules, providing valuable insights to users about which covariate spaces cause the model to perform poorly.
>
> > You do not mention the model you used to produce your results. (Or do you always use the same model for the main prediction and the error prediction, if yes, why?)
>
> We use the same class of model and hyper-parameters by default to avoid biasing the results by explicitly fine-tuning the error estimator. However, the objective is to find the best possible error estimator, and users can choose any model that suits their problems.
>
> > In which domain do X and Y reside?
>
> In our experiments, the domains of X and Y have been $\mathbb{R}^p $ and $\mathbb{R}$, with $p$ being the number of variables. The continuity or discreteness of the data is not important. It can be anything, as long as a bounded score value is attributed to each combination of prediction and true labels.
>
> > Which type of R² score is used here? What is the reference model? How is the R² score used in the production scenario, as it is batched based the application on sequential data points is not natively defined. Against your assumption in section 2, the R² score is not bounded! but be always smaller than 1. R² is only bounded if it is viewed as a data descriptor that assumes that a linear model is used to describe the data, then it is a measure for the explainable variance. This specification confuses more than it help. The R² formulation you use should be clearly specified in mathematical notation.
>
> We use the classical R² score from the scikit-learn library, defined as $\( R²(y, \hat{y}) = 1 - \frac{\sum (y - \hat{y})^2}{\sum (y - \bar{y})^2} \)$. However, this is not what we monitor in our experiments. Throughout the paper, we only use the R² score to describe the general performance of the error estimator. In production, the loss we monitor and update sequentially is the absolute residual $\( \ell(X, Y) = |y - \hat{y}| \)$. We apologize for the confusion and will make it clearer that the monitored loss function is the absolute distance $\( \ell(X, Y) \)$, computed with an output Y normalized between (0, 1).
>
> > If the Hoeffding interval is based on the Hoeffding Inequality, does your application fulfill all stated requirements?
>
> For both confidence sequences and intervals, the only assumption we need is the boundedness of the loss. In all our experiments, we ensure the score is bounded by using the absolute distance with bounded output.
>
> > What error metric is used for the stated experiment? Accuracy?
>
> We use the absolute distance between the predicted probability and the label.
>
> > General Question on DDM and PHT
>
> The main differences between these methods and ours are that they require access to ground truth labels to monitor performance, whereas our method does not.
>
> > Are you referring to joint probabilities or conditional probabilities? So, S(X)=1 and E>q or S(X)=1 given E>q?
>
> We were referring to the joint probability.
>
> > Domain knowledge regarding performance-based drift detection methods is not mentioned.
>
> We will include it along with the suggested references.

---

> > ### Comment · Reviewer_FDg9 · 2024-08-12
> >
> > Thank you for your detailed rebuttal! I'll briefly outline my remaining questions and concerns:
> >
> > ```
> > computed with an output Y normalized between (0, 1).
> > ```
> >
> > **MOST IMPORTANT QUESTION**: Please verify that this does not violate the independence assumptions of Hoeffding's Inequality. Based on my experience, having attempted to use Hoeffding's Inequality in regression for non-stationary environments, I can confirm that Hoeffding's Inequality requires both bounded and INDEPENDENT random variables. By normalizing your data targets, you may introduce interdependencies among them, which could be problematic.
> >
> > ```
> > The claims regarding the properties of “traditional” drift detection methods
> > ```
> > I agree with your point, but your reference specifically mentions tests based on distributions. There is an entire class of performance-based drift detectors that do not have these issues but do require target labels. Please clarify this distinction in your discussion.
> >
> > ```
> > In addition, the use of a reference window is presented as a weakness, but the proposed method also uses a calibration set, which can be seen as a reference window.
> > ```
> >
> > Unfortunately, this point was not addressed in your rebuttal.
> >
> > ```
> > You do not mention the model you used to produce your results.
> > ```
> >
> > While you mentioned that the model can be changed, what specific model did you use for your evaluations? This is important for reproducibility.
> >
> >
> > ```
> > General Question on DDM and PHT
> > ```
> > I understand that DDM and PHT require labels, while your approach does not. However, my question was about what happens afterward. After using your proxy model to predict the error, you still need to determine whether there is a drift. This decision-making process seemed quite similar to DDM to me. Could you elaborate on the differences, if any, or clarify if your method was inspired by DDM?

---

> ### Author Response · Authors · 2024-08-12
>
> Thank you for your response. Please find below our answers to the remaining points you raised:
>
> > Please verify that this does not violate the independence assumptions of Hoeffding's Inequality. Based on my experience, having attempted to use Hoeffding's Inequality in regression for non-stationary environments, I can confirm that Hoeffding's Inequality requires both bounded and INDEPENDENT random variables. By normalizing your data targets, you may introduce interdependencies among them, which could be problematic.
>
> You’re correct that Hoeffding’s Inequality, as well as the inequalities used for confidence sequences, require both bounded and independent random variables. Regarding the normalization, we are only rescaling the output before constructing or splitting the data into production, calibration, and training sets using min-max scaling with the same minimum and maximum values.  To the best of our knowledge, this rescaling process should not violate the assumption of independence.
>
>
> > I agree with your point, but your reference specifically mentions tests based on distributions. There is an entire class of performance-based drift detectors that do not have these issues but do require target labels. Please clarify this distinction in your discussion.
>
> In the second paragraph of the Introduction (L 25-32), where we describe methods that might detect harmful shifts without labels, we will also include techniques that do require labels in the final version of the paper, including the ones you suggested.
>
> >In addition, the use of a reference window is presented as a weakness, but the proposed method also uses a calibration set, which can be seen as a reference window.
>
> We agree that the calibration set can be seen as a window, but our comment was about inference time. Our method, being completely sequential, does not require defining a window for the production sample size, as it processes the data as it arrives, one by one. In Lines 17-20, we will specify that the prespecified sample set refers to the production set.
>
> > You do not mention the model you used to produce your results. While you mentioned that the model can be changed, what specific model did you use for your evaluations? This is important for reproducibility.
>
> For the image dataset, as stated in Lines 268-272, we used a ResNet-50 as the base model, modifying only the head for error estimation. For the tabular data, we consistently used a Random Forest regressor/classifier with the default parameters from the scikit-learn library (num_trees=100, max_depth=None, boostrap=True). We will clarify this further in the paper.
>
>
> >General Question on DDM and PHT. I understand that DDM and PHT require labels, while your approach does not. However, my question was about what happens afterward. After using your proxy model to predict the error, you still need to determine whether there is a drift. This decision-making process seemed quite similar to DDM to me. Could you elaborate on the differences, if any, or clarify if your method was inspired by DDM?
>
> Our paper is not inspired by DDM but by the work of Ramdas et al. (2022). While there may be some similarities in the decision-making process, as DDM computes an approximate confidence interval in the reference set (called the concept set) and compares it with a threshold value, there is a substantial difference. DDM is designed for contexts where the model also changes dynamically. Unlike DDM, the method proposed by Ramdas et al. (and by extension, our methods) uses a calibration step and defines the warning threshold based on statistical tools that provide provable false alarm guarantees in finite samples with minimal assumptions, such as independence and bounded loss.

---

### Official Review · Reviewer_KyAy · 2024-07-12

**Soundness:** 3
**Presentation:** 4
**Contribution:** 2
**Rating:** 7
**Confidence:** 4

**Summary:**

This paper proposes a new method for identifying harmful distribution shifts when no labels are available at test time. This work is a good contribution to the field of ML and has practical importance.

**Strengths:**

- The paper proposes a theoretically motivated method for the detection of harmful distribution shifts with no labels at test time;
- The paper discusses the conditions under which the method controls Type-I errors;

**Weaknesses:**

- In my view, there are two main weaknesses in this paper:
  1. The error prediction function is a function of the covariates X. That implies that this method only works in detecting harmful shifts that involve shifting the distribution of X (e.g., covariate shifts). If the shifts happen only in P_{Y|X}, instead, the method would have low power. However, this limitation is not discussed by the authors (Please let me know if I have any misunderstanding here). If this is true, the authors should compare their work with [1], because it's not clear how their work is different in nature from that work. If it turns out that both works have a similar nature, the new method would have to be superior in terms of performance;
  2. In the experiments, the authors do not compare their method with any other competitors. You could, for example, compare your method with covariate shift detection methods [1,2] (which do not rely on the presence of labels).

References:

[1] Ginsberg, Tom, Zhongyuan Liang, and Rahul G. Krishnan. "A learning based hypothesis test for harmful covariate shift." arXiv preprint arXiv:2212.02742 (2022).

[2] Polo, Felipe Maia, Rafael Izbicki, Evanildo Gomes Lacerda Jr, Juan Pablo Ibieta-Jimenez, and Renato Vicente. "A unified framework for dataset shift diagnostics." Information Sciences 649 (2023): 119612.

**Questions:**

The questions I have pertain to the first point I mentioned in the weaknesses section. Could you clarify that point for me? I would be willing to increase my score if I understand how your work builds on the previous research by Ginsberg et al. and recognize a significant difference.

**Limitations:**

Limitations are discussed.

---

> ### Author Rebuttal · Authors · 2024-08-04
>
> We thank the reviewer for taking the time to review our paper and for providing valuable references that we were not aware of. We appreciate the opportunity to clarify our contributions in light of these references.
>
> First, we note that paper [2] focuses on detecting distribution shifts in general and not specifically harmful shifts, unlike our method and the first suggested paper [1]. Consequently, using [2] as a benchmark in cases where there are harmful/benign shifts would be unfair since it does not differentiate between harmful and benign shifts and would likely result in a much higher rate of false alarms. Therefore, the first suggested paper is more relevant to our work as it directly targets harmful shifts.
>
> However, we must highlight that comparing our method directly with those in [1] is challenging due to fundamental differences in setup. The methods in [1] are designed for an offline setup, requiring a batch of production data to train their statistics and raise alarms for a batch of production data. In contrast, our methodology was designed for an online setup where shifts occur gradually and continuously, necessitating real-time decisions without observing an unlabelled batch of examples first. Our methodology aims to detect harmful performance shifts on the fly as observations are received one by one and do not require access to the full production data to fit the statistic.
>
> Additionally, applying offline methods in an online setting would be impractical and unfair, as these methods require learning a model or statistics from a batch of data. It would also be computationally expensive, as [1] requires learning a new model for each batch of production data. In this case, we would have to learn a number of models proportional to the size of the production data.
>
> However, we evaluated our method against the best method from [1] (Detectron) in a batch setting, increasing the size of production data or out-of-distribution (OOD) data. We generated the shifts similar to the setting described in Section 5.2 and ensured no shift in the first 1300 samples of the production data. We used NHANESI classification data as Detectron is only available for classification tasks. We replicated the experiments 50 times, resulting in a total of 10200 different shifts.
>
> Our results, summarized in the table below, show that for smaller sample sizes (100 and 1000), our method did not detect any shifts, which is expected since no shift occurred in the initial 1300 samples, while Detectron raised a significant number of alarms (1126 out of 1700 shifts for size 100 and 1493 for size 1000), all of which were false alarms. For larger sample sizes, while the method of [1] shows high power in detecting shifts, it also exhibits high false alarms. In contrast, our method demonstrates lower power but significantly better FDP control, aligning with our goal of minimizing false alarms.
>
> These results validate that our method performs robustly in a batch setting, as well as working effectively in the online setting considered in the paper.
>
> | OOD Size | Power Detectron | FDP Detectron        | Power SHSD | FDP SHSD |
> |----------|-----------------|----------------------|------------|----------|
> | 100      | N/A             | 1                    | N/A        | 0        |
> | 1000     | N/A             | 1                    | N/A        | 0        |
> | 2000     | 0.96            | 0.61                 | 0.40       | 0.02     |
> | 3000     | 0.98            | 0.60                 | 0.63       | 0.02     |
> | 3500     | 0.98            | 0.60                 | 0.67       | 0.02     |
> | 8593     | 0.98            | 0.60                 | 0.74       | 0.04     |
>
> We hope this clarifies the distinctions and the rationale behind our methodological choices. We will also clarify these differences and include additional experiments demonstrating them in the final version of the paper. Thank you again for your insightful feedback.

---

> > ### Comment · Reviewer_KyAy · 2024-08-09
> >
> > Thank you for your reply!
> >
> > I wanted to clarify some points:
> > 1. As far as I understand, in a practical sense, the main difference is that one method is designed for online settings while the other is designed for offline settings. Is that true?
> > 2. I did not understand if your method is designed to detect harmful shifts of X. Is that the case?
> > 3. Do you know why the Detectron has such a bad FDP in your experiment?

---

> ### Author Response · Authors · 2024-08-09
>
> Thank you for your response.
>
> > As far as I understand, in a practical sense, the main difference is that one method is designed for online settings while the other is designed for offline settings. Is that true? I did not understand if your method is designed to detect harmful shifts of X. Is that the case?
>
> Yes, the major difference is that our method is designed for online use, while the other is for offline. Regarding whether our method is designed to detect harmful "shifts of X" (i.e. covariate shifts), our central assumption is Assumption 4.1, i.e. that the selector function generalizes in terms of FDP from calibration to production. This is more likely to hold in the case of pure covariate shift, where the relationship Y | X is invariant. However, it may still be effective if this relationship changes, provided the model tends to have high error in the same regions of the input space before and after the shift. Note that in our empirical experiments, we study shifts in natural datasets, where the Y | X relationship is very likely to be at least somewhat non-invariant. Our method is shown be effective in these cases.
>
> > Do you know why the Detectron has such a bad FDP in your experiment?
>
> Detectron’s main idea is to learn a Disagreement Classifier that performs as well as the original model on the training distribution while disagreeing with the original/base model's predictions on the production data. This method is highly sensitive to the performance of the base classifier, the function class used, and the size and nature of the production data. While it effectively detects harmful shifts (as shown by its high power in our experiments), it may fail when the shift is benign.
>
> Below is a simple code example that generates a plot illustrating a failure case. In this example, the training data points is represented in red and green, with the ground truth shaded in corresponding colors. The base model is shown as a solid black line. For simplicity, we assume the model is a perfect classifier. The data has shifted to the right, creating unlabeled production data, all correctly classified by the base classifier.
>
> We’ve also depicted the potential learnable classifier as a dashed line, representing the boundary of all possible functions, which depends on the model type, complexity used for the disagreement classifier, and the nature and size of the data. We have shown a potential disagreement classifier in orange that performs similarly to the original model on training data but disagrees on the predictions of the base classifier in the production data. As shown, even with a benign shift, we can still find a disagreement classifier that performs well on training data but disagrees significantly in production, raising a false alarm.
>
> ```{python}
> import matplotlib.pyplot as plt
> import numpy as np
>
> # Generating sample data
> x = np.linspace(-3, 3, 500)
> base_classifier = 2 * np.sin(x)
> shaded_area = 2 * np.sin(x)
> boundary_upper = 5 * np.sin(x)
> boundary_lower = 0.5 * np.sin(x)
> disagreement_classifier = 3.5 * np.sin(x) * (x <= 0) + 4.5 * np.sin(x) * (x > 0)
>
> np.random.seed(0)
> x_class1 = np.random.uniform(-2, -1, 10)
> y_class1 = np.random.uniform(-2, 1, 10) + 3
>
> x_class2 = np.random.uniform(-2, -1, 10)
> y_class2 = np.random.uniform(-5, -2, 10) - 3
>
> x_class1_shifted = np.random.uniform(-2, -1, 20) + 3
> y_class1_shifted = np.random.uniform(-2, 2, 20) + 4.5
> x_class2_shifted = np.random.uniform(-2, -1, 10) + 3
> y_class2_shifted = np.random.uniform(-5, -2, 10) + 2
>
> x_unlabelled = np.concatenate([x_class1_shifted, x_class2_shifted])
> y_unlabelled = np.concatenate([y_class1_shifted, y_class2_shifted])
>
> # Plotting
> plt.figure(dpi=200, figsize=(8, 5))
>
> plt.fill_between(x, shaded_area, 12, color='green', alpha=0.1, edgecolor='none', label='Ground Truth (Class 1)')
> plt.fill_between(x, -12, shaded_area, color='red', alpha=0.1, edgecolor='none', label='Ground Truth (Class 2)')
>
> plt.scatter(x_class1, y_class1, c='green', marker='+', label='Training Data (Class 1)')
> plt.scatter(x_class2, y_class2, c='red', marker='+', label='Training Data (Class 2)')
>
> plt.scatter(x_unlabelled, y_unlabelled, facecolors='none', edgecolors='black', label='Unlabelled Production Data')
>
> plt.plot(x, boundary_upper, 'gray', linestyle='--', label='Set of Potential Learnable Classifiers')
> plt.plot(x, boundary_lower, 'gray', linestyle='--')
> plt.plot(x, base_classifier, 'k-', label='Base Classifier')
> plt.plot(x, disagreement_classifier, 'tab:orange', linestyle='-', label='Disagreement Classifier')
>
> plt.legend(loc='upper left', bbox_to_anchor=(1, 1), fontsize='small', frameon=False)
> plt.grid(False)
> plt.tight_layout()
> plt.axis('off')
> plt.show()
> ```

---

> > ### Comment · Reviewer_KyAy · 2024-08-09
> >
> > Thank you for the detailed explanation! Please include our discussion in your final version of the paper.
> >
> > I have raised my score.

---

> > > ### Author Response · Authors · 2024-08-13
> > >
> > > Thank you for your valuable feedback and for raising your score. We appreciate your insights and will incorporate our discussion into the final version of the paper.

---

### Official Review · Reviewer_4vCN · 2024-07-13

**Soundness:** 3
**Presentation:** 3
**Contribution:** 3
**Rating:** 6
**Confidence:** 3

**Summary:**

This work is an extension of Podkopaev and Ramdas. They propose a framework to detect the harmful distribution shift without accessing the true labels during detection. To do that, the authors introduce an error estimator model to measure the error scores. Besides, the authors propose a strategy to manage false positives by leveraging these imperfect error predictions. Experiments show that the method effectively controls false positives and performs well under various distribution shifts.

**Strengths:**

- The label-free solution is essential for practical applications.
- Algorithm-agnostic makes the approach more applicable.
- The approach does not rely on pre-assumptions about data distribution.
- The authors provide detailed mathematical derivations.

**Weaknesses:**

- In the experiments section, the authors only compare with a baseline scheme and ignore other existing distribution shift detection algorithms (e.g., [1-2]), although they may not claim to only detect harmful distribution shifts.

### Reference
1. Hinder F, Artelt A, Hammer B. Towards non-parametric drift detection via dynamic adapting window independence drift detection (dawidd)[C]//International Conference on Machine Learning. PMLR, 2020: 4249-4259.
2. Frias-Blanco I, del Campo-Ávila J, Ramos-Jimenez G, et al. Online and non-parametric drift detection methods based on Hoeffding’s bounds[J]. IEEE Transactions on Knowledge and Data Engineering, 2014, 27(3): 810-823.

**Questions:**

- In lines 117-118, could the authors explain how they connect the upper bound of the training dataset to the lower bound of the test dataset to derive this probability inequality?
- The proposed method does not rely on distribution assumptions and has very low accuracy requirements for error estimation (In section 4.1). Additionally,  there are no hyper-parameters that need to be preset (in section 4.2). However, there is no free lunch. Despite these excellent properties, does the method will face any other challenges?

**Limitations:**

- The proposed approach is to detect harmful distribution shifts in production environments, which is a very interesting problem setting. However, there are many out-of-distribution detection and robust learning algorithms that can cover or solve the harmful distribution shift problem, or even solve it directly, which makes the proposed method not so necessary in practical applications.

---

> ### Author Rebuttal · Authors · 2024-08-04
>
> We thank the reviewer for taking the time to review and acknowledge our work.
>
> > In the experiments section, the authors only compare with a baseline scheme and ignore other existing distribution shift detection algorithms (e.g., [1-2]), although they may not claim to only detect harmful distribution shifts.
>
> Regarding the suggested paper, neither would be a suitable baseline for our method as they either attempt to detect shifts in general or require access to labels to assess model performance. Comparing them with our method would be unfair since they are likely to have high false alarm rates in a context where there are both harmful and benign shifts. Moreover, the primary purpose of our methodology is to detect harmful performance shifts without using labels.
>
> However, the closest baseline to ours is the one suggested by Reviewer KyAy, although it is designed for batch or offline contexts rather than the sequential or online context of our work. We provide comparisons (please refer to our response to Reviewer KyAy) showing that our method also works in the batch context.
> >  However, there is no free lunch. Despite these excellent properties, does the method will face any other challenges?
>
> Regarding the limitations, we acknowledge that our method may perform less effectively when there is a significant concept shift or when we do not have enough calibration data to train and calibrate our proxy. We will include this consideration in the final version of the paper.
> > In lines 117-118, could the authors explain how they connect the upper bound of the training dataset to the lower bound of the test dataset to derive this probability inequality?
>
> Below, you will find the detailed derivation of Equation 7 (Lines 117-118):
>
> $$
>  P_{H_0}\left\[ \exists t \geq 1: \Phi_m(E_1, \dots, E_t) = 1 \right\]
> $$
> $$
> = P_{H_0}\left\[\exists t \geq 1: \hat{L}(E_1, \ldots, E_t) > \hat{U}(E_1^{0}, ..., E_1^{0})+ \epsilon_{\text{tol}}\right\]
> $$
> $$
> = P_{H_0}\left\[\exists t \geq 1:  \left(\hat{L}(E_1, \ldots, E_t) -\frac{1}{t} \sum_{k=1}^t \theta(P_{E}^{k})\right) - \left(\hat{U}(E_1^{0}, \ldots, E_n^{0})-\theta(P_{E}^{0})\right) >  \epsilon_{\text{tol}} - \left( \frac{1}{t} \sum_{k=1}^t \theta(P_{E}^{k}) - \theta(P_{E}^{0}) \right) \right\]
> $$
> $$
> \leq P_{H_0}\left\[\exists t \geq 1:  \left(\hat{L}(E_1, \ldots, E_t) -\frac{1}{t} \sum_{k=1}^t \theta(P_{E}^{k})\right) - \left(\hat{U}(E_1^{0}, \ldots, E_n^{0})-\theta(P_{E}^{0})\right) >  0 \right\]
> $$
> $$
> \leq P_{H_0}\left\[\exists t \geq 1:  \left(\hat{L}(E_1, \ldots, E_t) -\frac{1}{t} \sum_{k=1}^t \theta(P_{E}^{k})\right) >  0 \right\] + P_{H_0}\left\[\left(\hat{U}(E_1^{0}, \ldots, E_n^{0})-\theta(P_{E}^{0})\right) <  0 \right\]
> $$
> $$
> \leq \alpha_{\text{source}} + \alpha_{\text{prod}}
> $$
> The third last line is because we are under $H_0$ and the second last line is due to if $\exists t \geq 1:  \left(\hat{L}(E_1, \ldots, E_t) -\frac{1}{t} \sum_{k=1}^t \theta(P_{E}^{k})\right) - \left(\hat{U}(E_1^{0}, \ldots, E_n^{0})-\theta(P_{E}^{0})\right) >  0$, then either $\exists t \geq 1:  \hat{L}(E_1, \ldots, E_t) -\frac{1}{t} \sum_{k=1}^t \theta(P_{E}^{k}) >  0$ or $\hat{U}(E_1^{0}, \ldots, E_n^{0})-\theta(P_{E}^{0}) <  0$

---

> > ### Comment · Reviewer_4vCN · 2024-08-13
> >
> > Thank you for addressing the question, it is very clear to me.

---

### Decision · Program_Chairs · 2024-09-25

**Decision:**

Accept (poster)

**Comment:**

The paper introduces a method for detecting harmful shifts without having access to ground truth labels. The reviewers agree that the problem is well motivated, and the main contribution of the paper, allowing to handle the lack of access to ground truth labels, makes the contribution quite practical. The techniques are noted to be novel and insightful, as well as motivated from mathematical derivations. There were some issues raised regarding the comparison with existing works (see the baseline mentioned by KyAy and related works mentioned by 4vCN), but this was mitigated by the authors’ response, and the required modification to the paper is small enough to be achievable towards the final version. Concluding, the paper provides a novel solution to an important problem, and would be a welcome addition to Neurips.